# An Analytical Study of Utility Functions in Multi-Objective Reinforcement Learning

**Manel Rodriguez-Soto**
Artificial Intelligence
Research Institute (IIIA-CSIC)
Bellaterra, Spain
manel.rodriguez@iiia.csic.es

**Juan A. Rodriguez-Aguilar**
Artificial Intelligence
Research Institute (IIIA-CSIC)
Bellaterra, Spain
jar@iiia.csic.es

**Maite Lopez-Sanchez**
Universitat de Barcelona (UB)
Barcelona, Spain
maite_lopez@ub.edu

## Abstract

Multi-objective reinforcement learning (MORL) is an excellent framework for multi-objective sequential decision-making. MORL employs a utility function to aggregate multiple objectives into one that expresses a user's preferences. However, MORL still misses two crucial theoretical analyses of the properties of utility functions: (1) a characterisation of the utility functions for which an associated optimal policy exists, and (2) a characterisation of the types of preferences that can be expressed as utility functions. In this paper, we contribute to both theoretical analyses. As a result, we formally characterise the families of preferences and utility functions that MORL should focus on: those for which an optimal policy is guaranteed to exist. We expect our theoretical results to foster the development of novel MORL algorithms that exploit our theoretical findings.

## 1 Introduction

Sequential decision-making problems are ubiquitous, impacting areas like autonomous driving [6], robotics [35], finance [3] and healthcare [4], to name a few. Recently, Reinforcement Learning (RL) has emerged as a pivotal framework for addressing sequential decision-making tasks [11, 13]. Most of the RL literature has focused on problems for which an agent deals with a single objective (e.g. get rich in finance, win a race). However, real-world scenarios often present multiple, conflicting objectives [32] (e.g., a self-driving car must ensure safety, efficiency, and passenger comfort).

Multi-Objective Reinforcement Learning (MORL) has developed as one of the most promising frameworks for addressing multi-objective decision-making [19, 18, 22]. Despite its novelty compared to single-objective RL, the current state of the art in MORL shows promising results for tackling real-world problems that are inherently multi-objective [10, 32]. Most MORL approaches are *utility-based* and assume that there exists a *utility function* [10] that combines all objectives into a single one, allowing the learning agent to ponder between them. However, deciding the most appropriate utility function is a problem in itself. Given that, the literature on MORL focuses on learning a set of candidate policies, called the *undominated set* [10], which maximise all possible utility functions. In that way, once a utility function is decided, the decision-maker can directly select the policy from the undominated set that maximises it.

38th Conference on Neural Information Processing Systems (NeurIPS 2024).

Thus, utility functions are widely considered a fundamental concept of MORL [31]. Utility functions capture a user's preferences over different objectives and drive the learning [22]. Hence, both concepts (utilities and preferences) are at the core of state-of-the-art MORL. The state-of-art approach of considering the undominated set as the most general solution concept of MORL relies on two assumptions:

1. **On utilities:** It assumes that for every utility function, there exists a policy optimising it.
2. **On preferences:** It assumes that any user preference can be expressed as a utility function.

Unfortunately, none of the assumptions is correct. There are many examples of preferences that cannot be expressed with a utility function (e.g., the lexicographic order [34], as proved in [5]). Likewise, even for problems with a finite amount of possible states and actions, there are many utility functions for which there is no optimal policy for any state (we provide an explicit example in the later sections).

These two counterexamples raise the need for answering the following two main theoretical questions: (1) for which type of utility functions are optimal policies guaranteed to exist? (2) what types of preferences can be represented as utility functions? The state of the art on MORL has not addressed these fundamental research questions so far.

Against this background, we propose an in-depth analysis of utility functions in MORL by means of the following three contributions:

1. **We provide two novel MORL fundamental theoretical concepts**. We introduce the first formal definition of preferences between policies in MORL and the first formal definition of utility maximisation in MORL.
2. Given a utility function in MORL, **we characterise the sufficient conditions that guarantee the existence** of an optimal policy maximising it.
3. **We characterise under which conditions we can express preferences between policies as utility functions.** These are represented by a type of function called *quasi-representative* utility function, which preserves the most preferred policies as its maximal points.

We expect that our theoretical results will lead to novel MORL algorithms that can exploit the analytical properties of the utility functions introduced here.

The remainder of this paper is organised as follows. Section 2 provides the necessary background in multi-objective reinforcement learning. Then, Section 3 provides sufficient conditions to guarantee the existence of utility-maximising policies. Next Section 4 characterises the family of preferences that can be represented with utility functions. Thereafter, Section 5 presents the related work. Finally, Section 6 summarises our main theoretical findings and sets paths for future research.

## 2 Background

### 2.1 Single-objective reinforcement learning

In single-objective reinforcement learning (RL), sequential decision-making problems are formalised as *Markov decision process* (MDP) [11, 27]. An MDP represents an environment in which an agent is capable of repeatedly acting upon it to make it to transition it to a different state, and immediately receive a scalar reward (representing the agent's objective) after each action:

**Definition 1** (Markov Decision Process). *A (single-objective)[1] Markov Decision Process (MDP) is defined as a tuple $\langle \mathcal{S}, \mathcal{A}, R, T \rangle$ of two sets and two functions: the set of states $\mathcal{S}$, the set of actions $\mathcal{A}(s)$ available at each state $s$, the reward function $R : \mathcal{S} \times \mathcal{A} \times \mathcal{S} \to \mathbb{R}$, and the transition function $T : \mathcal{S} \times \mathcal{A} \times \mathcal{S} \to [0, 1]$ specifying the probability $T(s, a, s') = \mathbb{P}(s' \mid s, a)$ that the next state is $s'$ if an action $a$ is performed upon the state $s$.*

An agent's behaviour in an MDP is called a *policy* $\pi$. A policy $\pi(s, a)$ describes how likely an agent will perform action $a$ if the agent is currently in state $s$. The agent's objective is to learn the policy that accumulates the maximum sum of discounted rewards. Thus, to evaluate a given policy, we need

---

[1]Through the paper, we refer to a single-objective MDP simply as an MDP.

to compute the (expected) discounted sum of rewards that an agent obtains by following it. This operation is formalised by means the so-called *value function* $V : \mathcal{S} \to \mathbb{R}$, defined as:

$$V^\pi(s) \doteq \mathbb{E}[\sum_{k=0}^{\infty} \gamma^k R(S_{t+k}, A_{t+k}, S_{t+k+1}) \mid S_t = s, \pi], \tag{1}$$

where $\gamma \in [0, 1)$ is the discount factor, indicating how much we care about future rewards. Value functions allow us to partially order policies [27]. Hence, they allow to formalise the agent's objective as learning the policy that maximises the value function. This policy is defined as the *optimal* policy:

**Definition 2** (Optimal policy). *Given an MDP $\mathcal{M}$, its optimal policy $\pi_*$ is the policy that maximises the value function $V^\pi$. Formally:*

$$V^{\pi_*}(s) \geq V^\pi(s), \tag{2}$$

*for every state $s$ of the MDP $\mathcal{M}$, and every policy $\pi$ of $\mathcal{M}$.*

The optimal policy is the solution concept in single-objective RL. For any MDP with a finite state and action space, at least one optimal policy exists, which is also deterministic and stationary [7].

## 2.2 Multi-objective reinforcement learning

Multi-objective reinforcement learning (MORL) deals with environments in which an agent pursues multiple objectives simultaneously (for example, in a healthcare context, the health and the autonomy of a patient). Recall that in single-objective RL, the reward function represents the agent's objective. Thus, MORL considers environments with multiple reward functions, called *Multi-Objective Markov Decision Processes* [19, 10]. Formally:

**Definition 3** (Multi-Objective MDP). *An $n$-objective Markov Decision Process (MOMDP) is defined as a tuple $\langle \mathcal{S}, \mathcal{A}, \vec{R}, T \rangle$ where $\mathcal{S}$, $\mathcal{A}$ and $T$ are the same as in an MDP, and $\vec{R} : \mathcal{S} \times \mathcal{A} \times \mathcal{S} \to \mathbb{R}^n$ is a vector of reward functions, providing a reward function $R_i$ for each objective $i \in \{1, \ldots, n\}$.*

Policies in a MOMDP are evaluated by means of a value function *vector* $\vec{V}$, defined as the vector of all value functions per objective $\vec{V}(s) = (V_1(s), \ldots, V_n(s))$.

If not all objectives can be fully fulfilled simultaneously, the agent needs to prioritise between them. To represent an agent's preferences with respect to multiple objectives, most approaches in MORL assume that the value function of each objective can be aggregated into a single function. That way, the agent's goal becomes to maximise this aggregated value function. This aggregation is performed by means of a *utility function* $u$ (also called a *scalarisation function*)[18, 22]. In MORL, a utility function $u$ is defined as a function mapping the domain of all value functions (a subset of the real coordinate space $\mathbb{R}^n$) to the real space $\mathbb{R}$. With $u$, the agent's goal can be expressed as learning a policy that maximises the function $(u \circ \vec{V})(s) = u(\vec{V}(s))$. Formally[2]:

**Definition 4** (Utility function). *Let $\mathcal{M}$ be a MOMDP of $n$ objectives. Any function $u : \mathbb{R}^n \to \mathbb{R}$ is a utility function of $\mathcal{M}$.*

The family of linear utility functions is especially notable. Any linear utility function $l$ returns a weighted sum of value functions $(l \circ \vec{V})(s) = \vec{w} \cdot \vec{V}(s)$. For linear utility functions, the scalarised problem of maximising $l \circ \vec{V}$ can be solved with single-objective reinforcement learning algorithms[3].

While in single-objective RL there is a clear definition of the solution concept (a deterministic and stationary optimal policy), there is no equivalent for MORL. Instead, the utility function is typically assumed to be *unknown*, and that we only have minor assumptions about it (e.g., that it is linear or that it is monotonically increasing) [19, 22, 10]. With that in mind, the solution concept in MORL is to learn a set of *candidate policies* $\pi$, with each of them optimising a possible utility function $u$. The next Section explores typical solution concepts in MORL.

---

[2]The presented definition of the utility function follows the *Scalarised Expected Returns* (SER) criterion, which is by far the most popular one in the MORL literature [10]. We focus exclusively on the SER criterion.

[3]Because the linear utility function for $\vec{V}$ also induces a utility function for $\vec{R}$. Notice that $u(\vec{V}(s)) = \vec{w} \cdot \vec{V}(s) = \vec{w} \cdot \mathbb{E}[\sum_{t=0}^{\infty} \gamma^k \vec{R}_{t+k+1} \mid s] = \mathbb{E}[\sum_{t=0}^{\infty} \gamma^k \vec{w} \cdot \vec{R}_{t+k+1} \mid s]$.

## 2.3 Solution concepts of MOMDPs

The solution concepts in MORL depend on how much is assumed about the utility function. If nothing is assumed, the goal is to learn the set of maximal policies for *any* utility function. It is important to remark what it means for a policy to be *maximal for a utility function*. By far, the majority of the MORL community follow the so-called *state-independent* (SI) criterion to define optimality [19, 18, 22, 10]. Given a MOMDP, this criterion considers that a policy $\pi_*$ *maximises* a utility function if and only if it is maximal among the expectation of possible initial states (for some value function $\vec{V}$ and the random variable of possible initial states $S_0$). We denote this expectation with $\vec{V}_{SI}$:

$$\vec{V}_{SI}^{\pi} \doteq \mathbb{E}[\vec{V}^{\pi}(S_0)]. \tag{3}$$

Due to its simplicity, the *state-independent* (SI) criterion is widely used in MORL and RL in general. While it is generally innocuous in single-objective RL, it can generate contradictory policies in multi-objective RL, as we will show in Example 4 below.

Considering this *state-independent* criterion, all solution concepts for MOMDPs have been formalised exclusively for it. Thus, the state-of-the-art general solution, the *undominated set*, is defined as the set of policies that are maximal for at least one utility function. Formally [10]:

**Definition 5** (Undominated set). *Given a MOMDP $\mathcal{M}$, its* undominated set $U(\mathcal{M})$ *is defined as the set of policies for which there exists a utility function $u$ with a maximal scalarised value.*

$$U(\mathcal{M}) \doteq \{\pi \in \Pi(\mathcal{M}) \mid \exists u : \forall \pi' \in \Pi(\mathcal{M}), \ u(\vec{V}_{SI}^{\pi}) \geq u(\vec{V}_{SI}^{\pi'})\}, \tag{4}$$

*where $\Pi(\mathcal{M})$ is the set of all possible policies of an MOMDP $\mathcal{M}$.*

We recall that the definition of undominated set makes no assumption on the structure of the utility function. If we constrain it to be a linear function, then the solution concept becomes the *convex hull*. The convex hull of a MOMDP contains all policies that are maximal for at least one linear utility function (again, according to the SI criterion). Formally [10]:

**Definition 6** (Convex hull). *Given an MOMDP $\mathcal{M}$, its convex hull $CH(\mathcal{M})$ is the subset of policies $\pi_*$ that are optimal for some weight vector $\vec{w}$:*

$$CH(\mathcal{M}) \doteq \{\pi \in \Pi(\mathcal{M}) \mid \exists \vec{w} \in \mathbb{R}^n : \forall \pi' \in \Pi(\mathcal{M}), \ \vec{w} \cdot \vec{V}_{SI}^{\pi} \geq \vec{w} \cdot \vec{V}_{SI}^{\pi'}\}, \tag{5}$$

*where $\Pi(\mathcal{M})$ is the set of policies of $\mathcal{M}$.*

# 3 Utility optimal policies

Recall that the MORL literature defines solution concepts following the *state-independent* criterion. However, for a proper analysis of utility functions in MORL, we require more precise definitions considering each and every state of a MOMDP.

Furthermore, recall that, in single-objective MDPs, value functions impose a partial order over policies of an MDP [27]. However, thanks to Banach's fixed point theorem, we know that a deterministic and stationary optimal policy always exists for any finite MDP (and, thus, for every state) [7]. These theoretical properties become much weaker in multi-objective MDPs. In particular, the Banach fixed point theorem does not generalise even for finite MOMDPs. Thus, we may have finite MOMDPs for which no optimal policy exists for any state. These "more precarious" theoretical results motivate even more the need for studying the existence of optimal policies in a MOMDP at two different levels: one at a *single-state* level (i.e., considering a single state), and another one at the *all-states* level (considering all states).

We begin by defining *utility optimality* at the state level: a given policy $\pi$ is optimal at state $s$ with respect to a given utility function $u$ if and only if it obtains more scalarised discounted returns than any other policy at $s$. Formally:

**Definition 7** (utility optimal policy at a state). *Let $\mathcal{M}$ be a MOMDP with state set $\mathcal{S}$. Let $\Pi^{\mathcal{M}}$ be the set of policies of $\mathcal{M}$. Let $u$ be a utility function. Then, a policy $\pi_* \in \Pi^{\mathcal{M}}$ is optimal with respect to utility function $u$ at state $s \in \mathcal{S}$ if and only if:*

$$(u \circ \vec{V}^{\pi_*})(s) \geq (u \circ \vec{V}^{\pi})(s), \tag{6}$$

*for every policy $\pi \in \Pi^{\mathcal{M}}$. We say that $\pi_*$ is $\langle u, s \rangle$-optimal for short.*

**Example 1.** *Consider a MOMDP $\mathcal{M}$ with two states: an initial state $s_1$ and a terminal state $s_2$. An agent can perform two actions ($a_1$, $a_2$) in this environment, with rewards $\vec{R}(s_1, a_1) = (1, 0)$, $\vec{R}(s_1, a_2) = (0, 1)$ respectively. Consider the utility function $u(x, y) = x + sin(y)$. The deterministic policy $\pi(s_1) = a_1$ that obtains vectorial value $\vec{V}^\pi(s_1) = (1, 0)$ is clearly $\langle u, s_1 \rangle$-optimal since $sin(y) < y$ for any $y \in [0, 1]$.*

In the same vein as in single-objective RL, given a MOMDP, we define a policy as utility optimal at the *all-states* level (or simply utility optimal) as a utility optimal policy in every state in the MOMDP. Formally:

**Definition 8** (utility optimal policy). *Let $\mathcal{M}$ be a MOMDP with state set $\mathcal{S}$. Let $\Pi^\mathcal{M}$ be the set of policies of $\mathcal{M}$. Let $u$ be a utility function. Then, a policy $\pi_* \in \Pi^\mathcal{M}$ is* optimal with respect to utility function $u$ *if and only if:*

$$(u \circ \vec{V}^{\pi_*})(s) \geq (u \circ \vec{V}^\pi)(s), \tag{7}$$

*for every policy $\pi \in \Pi^\mathcal{M}$, and every state $s \in \mathcal{S}$. We say that $\pi_*$ is $u$-optimal for short.*

**Example 2.** *In the MOMDP in Example 1, policy $\pi(s_1) = a_1$ is $u$-optimal since there are only two states, and the second one is terminal.*

We know that a (deterministic) $u$-optimal policy always exists for any linear utility function, as shown in Section 2.2. However, this is not always the case for arbitrary utility functions. The following three examples illustrate conditions that are not enough to guarantee the existence of neither deterministic nor stochastic utility optimal policies in finite MOMDPs. These conditions are:

1. That the utility function is **monotonically increasing** (a family of utility functions specially studied in MORL [18, 10]. Example 3 illustrates how this condition is not enough to guarantee a deterministic $u$-optimal policy.

2. That the utility function is **strictly monotonically increasing**. Example 5 shows how this condition is not enough to guarantee a stochastic $\langle u, s \rangle$-optimal policy for any given state $s$.

3. That the utility function is *both* **strictly monotonically increasing** and **continuously differentiable**. Example 4 shows how even assuming both conditions there are utility functions without stochastic $u$-optimal policies.

In the following Example 3 we consider the *Chebyshev* function (also called *Tchebycheff*, a well-known utility function in MORL [17, 19, 18, 10]. The Chebyshev function returns more scalar value the nearest a given input value $x$ is to a *reference* value $\vec{r}$. Moreover, the Chebyshev function is also monotonically increasing [18].

**Example 3.** *Let $\epsilon > 0$ be a small real number, $\vec{r} \in \mathbb{R}^n$ a reference value, and $\vec{w} \in \mathbb{R}^n$ a weight vector such that each $w_i \geq 0$. The Chebyshev function $\psi_{\vec{r}, \epsilon, w} : \mathbb{R}^n \to \mathbb{R}$ is defined as [17]:*

$$\psi_{\vec{r}, \epsilon, \vec{w}}(x) \doteq -(\max_i w_i \cdot |r_i - x_i| + \epsilon \cdot \sum_i w_i \cdot |r_i - x_i|). \tag{8}$$

*Let $\mathcal{M}$ be a 2-objective deterministic MDP with three states $s_1$, $s_2$, and $s_3$ such that $s_3$ is the terminal state. Regarding actions, there is one possible action in $s_1$, which has associated rewards $\vec{R}(s_1, a_1) = (1, 0)$. Action $a_1$ transitions the state to $s_2$. Then, in state $s_2$, there are two possible actions with associated rewards $\vec{R}(s_2, a_2) = (2, 20)$ and $\vec{R}(s_2, a_3) = (3, 1)$. All actions in $s_2$ transition to terminal state $s_3$.*

*This environment has two possible deterministic policies. The first policy is $\pi_1(s_1) = a_1, \pi_1(s_2) = a_2$. This policy obtains values $\vec{V}^{\pi_1}(s_1) = (3, 20), \vec{V}^{\pi_1}(s_2) = (2, 20)$. The second policy is $\pi_2(s_1) = a_1, \pi_2(s_2) = a_3$. This policy obtains values $\vec{V}^{\pi_2}(s_1) = (4, 1), \vec{V}^{\pi_2}(s_2) = (3, 1)$. We select as reference point $\vec{r} = (3.5, 20)$, associated weights $\vec{w} = (1, 1/19)$, and $\epsilon = 0$. For this configuration we have: for policy $\pi_1$, $\psi(\vec{V}^{\pi_1}(s_1)) = -0.5$, $\psi(\vec{V}^{\pi_1}(s_2)) = -1.5$. For policy $\pi_2$, $\psi(\vec{V}^{\pi_2}(s_1)) = \psi(\vec{V}^{\pi_2}(s_2)) = -1$. Clearly, $\pi_1$ is the only deterministic $\langle \psi, s_1 \rangle$-optimal policy, while $\pi_2$ is the only deterministic $\langle \psi, s_2 \rangle$-optimal policy. Thus, no deterministic $\psi$-optimal policy exists.*

Example 3 showed that deterministic $u$-optimal policies do not necessarily exist in finite MOMDPs, a fact that was already known in the MORL literature [18]. But we can go one step further: the

next Example 4 shows that being finite is not enough for an MOMDP to guarantee the existence of stochastic $u$-optimal policies. Moreover, the utility function from Example 4 is both *strictly monotonically increasing and continuously differentiable*:

**Example 4.** *Consider the utility function $u(x,y) = \sqrt{x^2+1} + \frac{y}{20}$, which is strictly monotonically increasing for any $(x,y) \in \mathbb{R}^+ \times \mathbb{R}$, and continuously differentiable in $\mathbb{R}^2$. Let $\mathcal{M}$ be the same 2-objective deterministic MDP from Example 3.*

*This environment has the same two deterministic policies from Example 3. The first policy is $\pi_1(s_1) = a_1, \pi_1(s_2) = a_2$, which obtains scalarised values $u(\vec{V}^{\pi_1}(s_1)) \approx 4.16, u(\vec{V}^{\pi_1}(s_2)) \approx 3.24$. The second policy is $\pi_2(s_1) = a_1, \pi_2(s_2) = a_3$, which obtains scalarised values $u(\vec{V}^{\pi_2}(s_1)) \approx 4.17, u(\vec{V}^{\pi_2}(s_2)) \approx 3.21$.*

*It is easy to check that $\pi_1$ is the absolute $\langle u, s_2 \rangle$-optimal policy, while $\pi_2$ is the absolute $\langle u, s_1 \rangle$-optimal policy. We leave the details at Appendix A.1. Thus, no stochastic $u$-optimal policy exists.*

Our third example is a finite MOMDP and a strictly monotonically increasing utility function $u$ for which no stochastic $\langle u, s \rangle$-optimal policy exists for any state $s$.

**Example 5.** *Consider the utility function $u$ such that if $x = y$, then $u(x,x) = 0$, and otherwise $u(x,y) = \frac{1}{|x-y|}$. Let $\mathcal{M}$ be the 2-objective deterministic MDP from Example 1 with two states $s_1$ and $s_2$ such that $s_1$ is the initial state and $s_2$ is the terminal state. There are two possible actions in $s_1$ with associated rewards $\vec{R}(s_1, a_1) = (1, 0)$ and $\vec{R}(s_1, a_2) = (0, 1)$.*

*Every policy of $\mathcal{M}$ will be of the form $\pi(s_1, a_1) = p$ and $\pi(s_1, a_2) = 1 - p$ for some $p \in [0, 1]$. The vectorial value of such policy at state $s_1$ will be $\vec{V}^\pi(s_1) = (p, 1 - p)$. Notice how every possible value belongs to the Pareto Front of $\mathcal{M}$. Thus, any utility function is strictly monotonically increasing in $\mathcal{M}$, including the one defined in this example.*

*In particular, for any policy, its scalarised value will be $u(1, 1 - p) = \frac{1}{|2p-1|}$.*

*If for any policy $\pi$ we have $\pi(s_1, a_1) = p < \frac{1}{2}$, then the policy $\pi'$ such that $\pi(s_1, a_1) = p + \epsilon$, with $\epsilon > 0$ small enough so that $p + \epsilon < \frac{1}{2}$ obtains more scalarised value than $\pi$. Similarly, if $\pi(s_1, a_1) = p > \frac{1}{2}$, we can find an alternative policy $\pi'$ such that $\pi(s_1, a_1) = p - \epsilon$ that obtains more scalarised value than $\pi$. Thus, no $\langle u, s_1 \rangle$-optimal policy exists in this MOMDP.*

The result of Example 5 is specially significant because most MORL literature (with its state-independent criterion that only considers initial states $S_0$), focuses on computing $\langle u, S_0 \rangle$-optimal policies on strictly monotonically increasing utility functions [19, 10]. As we have shown in Example 5, such optimal policies are not guaranteed to exist.

Therefore, the logical next question after these three examples is to ask for which families of utility functions there exists at least one global utility optimal policy or at least one utility optimal policy for every state. In particular, we focus on stationary policies, like in single-objective RL. Formally:

**Problem 1.** *For which families of utility functions is guaranteed that a stationary $\langle u, s \rangle$-optimal policy will exist for every state $s$ of every possible finite MOMDP?*

**Problem 2.** *For which families of utility functions is guaranteed that a stationary $u$-optimal policy will exist for every possible finite MOMDP?*

Next, Sections 3.1 and 3.2 focus on providing *sufficient conditions* to guarantee the existence of utility optimal policies in a state and utility optimal policies in general, respectively.

## 3.1 Utility optimal policy at a state existence

This Section introduces a family of utility functions that solve Problem 1. In particular, we offer a sufficient condition to guarantee the existence of a stationary $\langle u, s \rangle$-optimal policy for every state $s$ of a finite MOMDP. This sufficient condition is that the utility function is continuous. Formally:

**Theorem 1.** *Let $\mathcal{M}$ be a finite MOMDP. Let $u$ be a continuous utility function for all value functions of all policies $\Pi(\mathcal{M})$ of $\mathcal{M}$. Then, for every state $s$ of $\mathcal{M}$, at least one stationary $\langle u, s \rangle$-optimal policy exists.*

*Proof 1.* See Appendix A.3. $\qquad\qquad\square$

Continuous utility functions are one the most extensively studied and applied family of functions due to their *well-behaved* properties (e.g., existence of absolute maximum and minimum). Nevertheless, recall that Theorem 1 only provides sufficient conditions, and thus there might exist $\langle u, s \rangle$-optimal policies for discontinuous utility functions. We offer such an example in the proof of Theorem 3.

### 3.2  Utility optimal policy existence

Demanding that the same policy is $u$-optimal for some utility function $u$ for every state of the MOMDP is a much harder problem than demanding it for a given state. Thus, in this case, it is not enough that the utility function is continuously differentiable (i.e., continuous and all partial derivatives also continuous), and it is not enough that the utility function is also strictly monotonically increasing (as seen in Example 4).

It is already known that, for linear utility functions, we can obtain a $u$-optimal policy. So, the question is if we can find at least another family of utility functions for which a $u$-optimal policy exists. Theorem 2 presents such a family: utility functions that result from composing an affine function together with a strictly monotonically increasing function. Formally:

**Theorem 2.** *Let $\mathcal{M}$ be a finite multi-objective MDP $\mathcal{M}$. Let $u$ be a utility function decomposable as $u(x) = h(g(x))$, with $g(x) : \mathbb{R}^n \to \mathbb{R}$ being an affine function, and $h(x) : \mathbb{R} \to \mathbb{R}$ being a strictly monotonically increasing function for all value functions of all policies $\Pi(\mathcal{M})$ of $\mathcal{M}$. At least one deterministic and stationary $u$-optimal policy exists.*

*Proof 2.* See Appendix A.4. □

Notice that, in particular, Theorem 2 also covers linear utility functions. Linear utility functions are one of the most widely applied families of utility functions in MORL [18, 10]. To finish this Section, we show an example of a function composed by an affine and a strictly monotonically increasing function (that hence satisfies Theorem 2) that produces a non-linear (and non-affine) utility function for which a $u$-optimal policy exists.

**Example 6.** *Consider any 2-objective MDP $\mathcal{M}$ where all rewards are positive (i.e., $\vec{R}(s,a) \in \mathbb{R}^+ \times \mathbb{R}^+$ for all $s, a$), and a utility function $u$ defined as*

$$u(x, y) = -\frac{1}{x + y + 3 + sin(x + y + 3)}. \tag{9}$$

*We decompose $u(x, y)$ as $u(x, y) = h(g(x, y))$ with $g(x, y) = x + y + 3$ being affine, and $h(x) = -\frac{1}{x + sin(x)}$ being strictly monotonically increasing. By Theorem 2, a $u$-optimal policy exists.*

Notice that Theorem 2 only provides sufficient conditions of utility functions $u$ for guaranteeing the existence of $u$-optimal policies. In fact, Example 2 shows a non-affine utility function for which an $u$-optimal policy exists in a particular MOMDP.

## 4  Preference relations in MORL

In the previous section, we characterised the utility functions for which we can compute a utility optimal policy. However, as mentioned in the Introduction, a more fundamental question remains unanswered: Which user's preferences can be expressed as utility functions in a given MOMDP?

We require formalising preference relations and their maximal elements in MOMDPs to answer this last question. Preference relations, also known as binary relations, allow us to express, among two elements of a set, which one we prefer [25, 14]. While the state of the art in MORL makes no distinction between preference relations and utility functions [10], it is important to maintain them as two separate concepts. First of all, let us provide a formal definition of preference relations in MORL, inspired by [25]:

**Definition 9** (Preference relation in a MOMDP). *Let $\mathcal{M}$ be a MOMDP of $n$ objectives. We define a preference relation in $\mathcal{M}$ as any binary relation $\succeq$ over at least one pair of value vectors of $\mathbb{R}^n$. In particular, we say that:*

- *a value function $\vec{V}_1 \in \mathcal{V}$ is* weakly preferred *to another value function $\vec{V}_2 \in \mathcal{V}$ if and only if $\vec{V}_1(s) \succeq \vec{V}_2(s)$ for every state $s$ of $\mathcal{M}$. In short, we denote $\vec{V}_1 \succeq_\mathcal{M} \vec{V}_2$.*

- *a value function $\vec{V}_1 \in \mathcal{V}$ is* strictly preferred *to another value function $\vec{V}_2 \in \mathcal{V}$ if and only if $\vec{V}_1(s) \succeq \vec{V}_2(s)$ for every state $s$ of $\mathcal{M}$ and not $\vec{V}_2(s') \succeq \vec{V}_1(s')$ for at least one state $s'$ of $\mathcal{M}$. In short, we denote $\vec{V}_1 \succ_\mathcal{M} \vec{V}_2$.*

If for two value vectors we have that $\vec{V}_1(s) \succeq \vec{V}_2(s)$ and $\vec{V}_2 \succeq \vec{V}_1(s)$, we say that they are indifferent, and we denote it with the $\approx$ symbol. Notice that this definition makes no assumption over the preference relation. We do not impose that this preference relation is a pre-order, a partial order, or a total order [9]. Considering that the MORL literature applies utility functions of all kinds, we did not want to restrict our definition.

Following the game theory literature, humans have preferences, and we (sometimes) can represent them as utility functions, but not the other way around [14]. In fact, sometimes, a utility function that fully represents our preferences may not exist. If such a utility function exists, we call it the *representative* utility function of preference relation $\succeq$. Formally:

**Definition 10** (Representative utility function). *Let $\mathcal{M}$ be a MOMDP, and let $\succeq$ be a preference relation in $\mathcal{M}$. Then, we define a utility function $u$ as* representative *of the preference relation $\succeq$ if and only if, for every pair of possible value functions $\vec{V}_1$, $\vec{V}_2$, and every state $s$ of $\mathcal{M}$:*

$$\vec{V}_1(s) \succeq \vec{V}_2(s) \iff (u \circ \vec{V}_1)(s) \geq (u \circ \vec{V}_2)(s). \tag{10}$$

Some (but not all) preference relations have representative utility functions. However, any utility function $u$, is representative of some preference relation $\succeq_u$ defined as exactly fulfilling Equation 10.

In order theory, for any quasi-order (i.e., a preference relation that is at least reflexive and transitive), we can define the concept of maximal elements [9]. In our case, given a preference relation $\succeq$ between the value functions of a MOMDP, its maximal elements would be the value functions associated with the policy that we expect the agent to learn. Formally:

**Definition 11** (Maximal element). *Let $\mathcal{M}$ be a MOMDP. Let $\succeq$ be a preference relation in $\mathcal{M}$ that is at least reflexive and transitive (a quasi-order). Then, the value vector $\vec{V}_*(s)$ of value function $\vec{V}_*$ is a maximal element in state $s$ if and only if for every other possible value function $\vec{V}$ of $\mathcal{M}$:*

$$\vec{V}(s) \succeq \vec{V}_*(s) \implies \vec{V}_*(s) \succeq \vec{V}(s). \tag{11}$$

**Example 7.** *In finite single-objective MDPs, the optimal value $V_*(s)$ is a maximal element for every state $s$, for the preference relation $\succeq$ defined as $V(s) \succeq V'(s) \iff V(s) \geq V'(s)$.*

As mentioned in the introduction above, not all preference orders in MORL can be represented as a utility function. One of the most well-known cases is the *lexicographic order* [5, 2]. Although Lexicographic MORL has been studied in detail [8, 29, 12, 30, 24], almost no work in MORL (with the exception of [23]) has noticed that an associated utility function does not exist in general. Let us see through an example why the lexicographic order cannot be represented as a linear utility function. For utility functions in general, we refer to Corollary 2 of [23].

**Example 8.** *Consider a MOMDP $\mathcal{M}$ with two states: an initial state $s_1$ and a terminal state $s_2$. An agent can perform three actions in this environment ($a_1, a_2, a_3$), with rewards $\vec{R}(s_1, a_1) = (1, 0)$, $\vec{R}(s_1, a_2) = (0, 1)$, and $\vec{R}(s_1, a_3) = (1, 1)$, respectively. Consider now the lexicographic order $\succeq$ such that objective 1 is always preferred to objective 2. Hence, $\vec{R}(s_1, a_3) \succ \vec{R}(s_1, a_1) \succ \vec{R}(s_1, a_2)$. Any linear utility function here will be of the form $u_{\alpha,\beta}(x, y) = \alpha \cdot x + \beta \cdot y$. For $u_w$ to represent $\succ$, it must satisfy $u_{\alpha,\beta}(1, 1) > u_{\alpha,\beta}(1, 0)$ and $u_{\alpha,\beta}(1, 0) > u_{\alpha,\beta}(0, 1)$, for example, $u_{10,1}(x, y) = 10x + y$.*

*However, policies can be stochastic, and thus, we can have for instance any policy $\pi$ such that $\pi(s_1, a_1) = p, \pi(s_1, a_2) = 1 - p$, with $1 \geq p \geq 0$, which has associated value $\vec{V}^\pi(s_1) = (p, 1 - p)$. Hence, the utility function must also satisfy $u(1, 0) > u(0.9, 0.1) > \cdots > u(0.1, 0.9) > u(0, 1)$. And it needs to be absolutely precise: $u(p + \epsilon, 1 - p - \epsilon) > u(p, 1 - p)$ for every $\epsilon > 0$ arbitrarily small. Thus, it is impossible to represent the lexicographic order as a linear utility function.*

While lexicographic orders cannot be represented with utility functions in MOMDPs, they do have maximal elements among finite MOMDPs. A utility function that shares the exact same maximal elements as a lexicographic order would be very helpful. With such a utility function, we could still find the policies that maximise a lexicographic order with state-of-the-art MORL algorithms. Having formalised maximal elements in MOMDPs for any quasi-order, we can introduce utility functions that *at least* preserve maximal elements. We call this kind of utility function *quasi-representative*. Formally:

**Definition 12.** *Let $\mathcal{M}$ be an MOMDP. Let $\succeq$ be a preference relation $\succeq$ in $\mathcal{M}$ that is at least reflexive and transitive (a quasi-order). Let $u$ be a utility function such that for every state $s$ of $\mathcal{M}$:*

$$\vec{V}_*(s) \text{ is a maximal element of } \succeq \text{ at state } s \iff \vec{V}_*(s) \in \arg\max_{\vec{V}}[(u \circ \vec{V})(s)]. \qquad (12)$$

*Then, we say that $u$ is* quasi-representative *of $\succeq$ in $\mathcal{M}$.*

**Example 9.** *Consider the MOMDP $\mathcal{M}$ in Example 8. The utility function $u(x, y) = 10x + y$ is quasi-representative of the lexicographic order because $u(1, 1) > u(1, 0)$ and $u(1, 1) > u(0, 1)$.*

In fact, quasi-representative utility functions allow us to define an equivalence relation between utility functions. Hence, by abuse of notation, we will also say that two utility functions are *quasi-representative* for a given MOMDP if and only if they share the same maximum elements for every state $s$ of this MOMDP.

**Example 10.** *Consider the same MOMDP $\mathcal{M}$ and the lexicographic order $\succeq$ from Example 8. For example, utility functions $u(x, y) = 10x + y$ and $u'(x, y) = 15x + y + 30$ share the same utility optimal policy (which is $\pi(s_1) = a_3$), and hence are quasi-representative for $\mathcal{M}$ and $\succeq$.*

Notice that if a utility function $u$ is representative of some preference relation $\succeq$, it is also quasi-representative of $\succeq$. Notice also that a utility function may be representative or quasi-representative of a given preference order for some MOMDP but not for other MOMDPs.

Now, given a MOMDP, what conditions must a preference order meet to be represented by a quasi-representative utility function? Essentially, it is sufficient to have a maximal element for every state of the MOMDP. We present now a family of preference orders for which a quasi-representative utility function always exists for every finite MOMDP:

**Theorem 3.** *Let $\succeq$ be a preference relation and $\mathcal{M}$ any finite MOMDP. Assume that $\succeq$ is: (1) complete (either $a \succeq b$ or $b \succeq a$ or $a \approx b$ for every two possible value vectors $a, b$ of $\mathcal{M}$); (2) transitive (if $a \succeq b$, and $b \succeq c$, then $a \succeq c$); and (3) at least one maximal element $\vec{V}(s)$ exists for every state $s$ of $\mathcal{M}$. Then, a quasi-representative utility function exists for $\succeq$ in $\mathcal{M}$.*

*Proof 3.* We offer a constructive proof. For every state $s$, consider its set of maximal elements $\vec{\mathcal{V}}_*(s)$ according to $\succeq$, which is non-empty for every state due to Condition (3). Since the preference relation is complete, for every state $s$ all its maximal elements will share the same value (i.e., $\vec{V}_1(s) = \vec{V}_2(s)$ for every two $\vec{V}_1, \vec{V}_2 \in \vec{\mathcal{V}}_*(s)$). Hence, without loss of generality, we consider that there is a single maximal element per state. Then, the number of maximal elements is at most $|S|$, and we can order them according to $\succeq$ (we can order them because $\succeq$ is total and transitive). Then, set a number between 1 and $|S|$ for each of these elements, ordered by $\succeq$. For every other vector $x \in \mathbb{R}^n$, set $u(x) = 0$. Now, by construction, for every state $s$ we have that $\max_{\vec{V}}(u \circ \vec{V})(s) \in \vec{\mathcal{V}}_*(s)$. In other words, $u$ is a quasi-representative utility function, such that it returns the most preferred value vector for each state $s$ according to $\succeq$. $\qquad\square$

Completeness and transitivity are very common conditions for preference relations in game theory [14]. The third condition is required for MOMDPs since we are dealing with an infinite amount of policies. For instance, Example 5 showed a finite MOMDP for which there is no maximum element for any environment state.

The main takeaway from Theorem 3 is that MORL algorithms should focus on the family of preference relations that fulfils all its conditions. Such conditions are sufficient to guarantee the existence of a quasi-representative utility function, as we just proved.

The family of preference relations satisfying the conditions of Theorem 3 has examples aplenty, such as the previously mentioned family of lexicographic orders. Moreover, any continuous utility function is representative of a total order that satisfies all conditions of Theorem 3.

# 5 Related work

Most of the literature in MORL focuses on creating novel solution concepts in MORL and algorithmic methods to solve them (e.g., [28, 33, 20, 24, 21]). Instead, we focus on characterising for which families of utility functions these solutions exist, a largely overlooked theoretical problem despite its relevant implications. Take for instance the work in [33], where Van Moffaert *et al.* present a method for computing the Pareto front of a given MOMDP. They implicitly assume that this Pareto front will always include the *solution* policy (a $u\rangle$-optimal policy in our terms), but as we have proven in Example 4, this is not always the case.

Then, regarding the study of preference relations in MORL, to the best of our knowledge the only other works in the literature apart from ours are [23, 26]. Skalse *et al.*'s theoretical results in [23] complement ours by stating that, for every so-called *objective* (a preorder between policies), a $u$-optimal policy exists if and only if this objective can be represented with a linear utility function. This aligns with our results in Theorem 2. However, they do not establish whether there may be more families of utility functions for which a $u$-optimal policy exists, as we do with Theorem 2. Moreover, our *preference* definition allows for ordering policies in each state of the environment, providing more granularity than their *objective* definition. This difference is also significant, because it allows us to identify issues in the solution concepts of MORL as we have tried to illustrate with Examples 5 and 4.

Next, Subramani *et al.* in [26] follow on the work in [23], but they tackle a different problem than us. Their focus is on compare the expressivity of the MORL framework with other frameworks. They aim to know which *objectives* (defined identically to [23]) can be represented on each framework. However, like [23], their *objective* definition does not allow them to order policies differently per state, unlike our *preference* definition.

To finish, closely related to our work, Miura in [15] tackles the problem of characterising preferences and their properties in constrained MDPs [1]. They define preferences as sets of *acceptable policies* and aim to find for which environments they can set the constraints and reward functions of a constrained MDP (CMDP) for which the acceptable policies are optimal. While CMDPs and MOMDPs share many similarities, they belong to separate research area. The major difference between them is that a CMDP has constraints, while an MOMDP has a utility function.

# 6 Conclusions

Multi-objective reinforcement learning (MORL) is the most promising framework for dealing with sequential decision-making problems with multiple objectives. In MORL, the learning agent ponders between the multiple objectives by means of a utility function aligned with the user's preferences. However, the state of the art in MORL has disregarded two fundamental theoretical problems related to utility functions: (1) for which utility functions an associated optimal policy is guaranteed to exist? and (2) which preference relations can be expressed as a utility function?

In this paper, we contributed to the state of the art in MORL by formalising both problems for the first time and by analysing each one. For utility functions, we first formalised the concept of *utility optimality* in MORL. Then, we provided sufficient and insufficient conditions for such a policy to exist for any finite MOMDP. For preference relations, we first formalise them for MOMDPs, and we also provide the minimal conditions to guarantee that they can be expressed as a particular type of utility function, the so-called *quasi-representative* utility functions. We expect our theoretical contributions to spark interest in both theoretical and practical MORL research. In fact, our results have direct practical consequences: to avoid contradictory policies, the MORL community needs to design algorithms that check that their learned policies are utility optimal.

We envision many directions for future research. On the theoretical side, a generalisation of the presented theoretical results to multi-agent multi-objective environments would be of great interest in the MORL literature [22]. On the algorithmic side, we expect to see the development of algorithms that exploit our Theorems to compute utility optimal policies.

## Acknowledgements

The research presented in this paper was supported by the EU-funded VALAWAI (# 101070930) project, the Spanish-funded VAE (# TED2021-131295B-C31) and Rhymas (# PID2020-113594RB-100) projects. This work was supported by grant PID2022-136787NB-I00 funded by MCIN/AEI/10.13039/501100011033. It was also funded by GUARDEN (101060693), Fairtrans (PID2021-124361OB-C33), AUTODEMO (SR21-00329), TAILOR (H2020-952215), PrepParticip2.0 ( 24S03545-001), grants 2021 SGR 00313 and 2021 SGR 00754. Maite Lopez-Sanchez belongs to the WAI research group (University of Barcelona) associated unit to CSIC by IIIA.

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

# A   Appendix

Next, we provide the proofs of all theoretical results of *An Analytical Study of Utility Functions in Multi-Objective Reinforcement Learning* that could not fit in the main paper.

## A.1   Last part of Example 4

**Example 11.** *Recall the utility function $u(x, y) = \sqrt{x^2 + 1} + \frac{y}{20}$.*

*As previously mentioned This environment has two possible deterministic policies. The first policy is $\pi_1(s_1) = a_1, \pi_1(s_2) = a_2$. This policy obtains values $\vec{V}^{\pi_1}(s_1) = (3, 20), \vec{V}^{\pi_1}(s_2) = (2, 20)$, and scalarised values $u(\vec{V}^{\pi_1}(s_1)) \approx 4.16, u(\vec{V}^{\pi_1}(s_2)) \approx 3.24$.*

*The second policy is $\pi_2(s_1) = a_1, \pi_2(s_2) = a_3$. This policy obtains values $\vec{V}^{\pi_2}(s_1) = (4, 1), \vec{V}^{\pi_2}(s_2) = (3, 1)$, and scalarised values $u(\vec{V}^{\pi_2}(s_1)) \approx 4.17, u(\vec{V}^{\pi_2}(s_2)) \approx 3.21$.*

*Any stochastic policy $\pi$ will be of the form $p\pi_1 + (1 - p)\pi_2$ with $1 \geq p \geq 0$. That means that:*

- *At state $s_1$ it will obtain value $\vec{V}^{\pi}(s_1) = (3p + 4(1 - p), 20p + (1 - p)) = (4 - p, 19p + 1)$, and scalarised value $u(\vec{V}^{\pi}(s_1)) = \sqrt{(4 - p)^2 + 1} + \frac{19p + 1}{20}$.*

- *At state $s_2$ it will obtain value $\vec{V}^{\pi}(s_2) = (2p + 3(1 - p), 20p + (1 - p)) = (3 - p, 19p + 1)$, and scalarised value $u(\vec{V}^{\pi}(s_2)) = \sqrt{(3 - p)^2 + 1} + \frac{19p + 1}{20}$.*

*Consider now the scalarised value of the stochastic policy $\pi$ as a function $u'(p, s)$ depending on the real variable $p$. Its derivative is*

$$u'(p, s) = \frac{p - \alpha_s}{\sqrt{p^2 - 2\alpha_s p + \alpha_s^2 + 1}} + \frac{19}{20}, \tag{13}$$

*where $\alpha_{s_1} = 4$, $\alpha_{s_2} = 3$. The derivative $u'(p, s_1)$ has a root $r_1 \approx 0.958$, and the derivative $u'(p, s_2)$ has a root $r_2 \approx -0.042$. Both roots are global minima, and thus $u'(0, s_1)$ is a global maximum for $[0, 1]$ at state $s_1$, and $u'(1, s_2)$ is a global maximum for $[0, 1]$ at state $s_2$. In other words, $\pi_1$ is the absolute $\langle u, s_2 \rangle$-optimal policy, while $\pi_2$ is the absolute $\langle u, s_1 \rangle$-optimal policy. Thus, no stochastic $u$-optimal policy exists.*

## A.2   Preliminaries for Theorems

This Section is devoted to prove an indispensable theorem: that our search for the utility optimal policy can be reduced to searching only among stationary policies.

**Lemma 4.** *Let $\mathcal{M}$ be any finite MOMDP. Then, for every policy $\pi$, there exists another stationary policy $\pi'$ such that, for every state $s$ of $\mathcal{M}$, it obtains the same expected returns: $\vec{V}^{\pi}(s) = \vec{V}^{\pi'}(s)$ for every state $s$.*

*Proof 4.* Direct generalisation from single-objective MDPs, in which for any policy $\pi$, there is another stationary policy $\pi'$ such that for every state $s_0$ it achieves the same value $V^{\pi}(s_0) = V^{\pi'}(s_0)$. See Proposition 1.1. of [16] for a full proof for single-objective MDPs. $\square$

**Theorem 5.** *Let $\mathcal{M}$ be any finite MOMDP. Let $u$ be a utility function under the SER criterion. Then:*

- *If an $\langle u, s \rangle$-optimal policy $\pi_*$ exists for a state $s$ of $\mathcal{M}$, there is at least another stationary policy $\pi'_*$ such that $\pi'_*$ is also $\langle u, s \rangle$-optimal.*

- *If an $u$-optimal policy $\pi_*$ exists for $\mathcal{M}$, there is at least another stationary policy $\pi'_*$ such that $\pi'_*$ is also $u$-optimal.*

*Proof 5.* We only cover the first case, with the second one being analogous. Let $\pi_*$ be such that $u(\vec{V}^{\pi_*})(s) \geq u(\vec{V}^{\pi})(s)$ for every state $s$. Then, by Lemma 4, there exists another stationary policy $\pi'_*$ such that $\vec{V}^{\pi_*}(s) = \vec{V}^{\pi'_*}(s)$ for every state $s$. Thus, $u(\vec{V}^{\pi'_*}(s)) = u(\vec{V}^{\pi_*}(s)) \geq u(\vec{V}^{\pi}(s))$, and so $\pi'_*$ is also $u$-optimal. $\square$

Notice that Theorem 5 need not be true for utility functions under the ESR criterion.

## A.3 Proof of Theorem 1

**Theorem 6.** *Let $\mathcal{M}$ be a finite MOMDP. Let $u$ be a continuous utility function for all value functions of all policies $\Pi(\mathcal{M})$ of $\mathcal{M}$. Then, for every state $s$ of $\mathcal{M}$, at least one stationary $\langle u, s \rangle$-optimal policy exists.*

*Proof 6.* Without loss of generalisation we only consider stationary policies thanks to Theorem 5.

Given $\mathcal{M}$, consider the polytope (i.e., $n$-dimensional bounded polyhedron) formed by a convex coverage set $CCS$ of $\mathcal{M}$ at state $s$ (which has a finite amount of deterministic stationary policies as vertices) [18].

Such polytope, by definition of $CCS$, envelops all images of all possible value functions at state $s$ for $\mathcal{M}$. Moreover, the image of any value function at state $s$ can be expressed as a convex combination of the deterministic stationary policies of $CCS$ at that state [18].

Since $u$ is a continuous function, and the polytope formed by $CCS$ is closed and bounded, by the Extreme Value Theorem there exists a maximum value vector $\vec{V}_*(s)$ for $u$ in the polytope.

Since for any value vector $\vec{V}_*(s)$ in the polytope there is an associated stochastic stationary policy [18], we can find the policy $\pi_*$ associated with $\vec{V}_*(s)$ that achieves the maximum value in the polytope.

Thus, there exists a $\langle u, s \rangle$-optimal policy for every state $s$ of the MOMDP. $\qquad\square$

## A.4 Proof of Theorem 2

**Lemma 7.** *For every finite single-objective MDP $\mathcal{M}$, any utility function $u$ that is strictly monotonically increasing preserves the ordering between policies. That is, for every two value functions $V_1$ and $V_2$, for every state $s$:*

$$(u \circ V_1)(s) > (u \circ V_2)(s) \iff V_1(s) > V_2(s), \tag{14}$$
$$(u \circ V_1)(s) = (u \circ V_2)(s) \iff V_1(s) = V_2(s). \tag{15}$$

*In particular, any optimal policy is also $u$-optimal in $\mathcal{M}$ and vice-versa.*

*Proof 7.* Direct from the definition of strictly monotonic function. $\qquad\square$

**Lemma 8.** *For every finite single-objective MDP $\mathcal{M}$, for any affine utility function $u$, there exists a deterministic and stationary $u$-optimal policy in $\mathcal{M}$.*

*Proof 8.* Any affine function $u$ is quasi-representative of another linear utility function $l$ defined as $l(x) \doteq f(x) - f(0)$. For any linear utility function $l$, there always exists a deterministic and stationary $l$-optimal policy. $\qquad\square$

With these two Lemmas, we can prove that there exists a family of non-linear utility functions for which an $u$-optimal policy exists (and moreover, the policy is deterministic and stationary): utility functions product of composing an affine function together with a strictly montonically increasing function.

**Theorem 9.** *Let $\mathcal{M}$ be a finite multi-objective MDP $\mathcal{M}$. Let $u$ be a utility function decomposable as $u(x) = h(g(x))$, with $g(x) : \mathbb{R}^n \to \mathbb{R}$ being an affine function, and $h(x) : \mathbb{R} \to \mathbb{R}$ being a strictly monotonically increasing function for all value functions of all policies $\Pi(\mathcal{M})$ of $\mathcal{M}$. At least one deterministic and stationary $u$-optimal policy exists.*

*Proof 9.* Direct consequence of combining Lemma 7 and Lemma 8.

We divide the proof in two steps. First, we prove that a function decomposable in a linear function and a strictly monotonically increasing function has $u$-optimal policies:

(i) First, as Lemma 7 states, applying a strictly monotonically increasing utility function to a single-objective MDP does not modify its set of deterministic and stationary optimal policies.

(ii) Second, every linear utility function $lu$ can transform a MOMDP $\mathcal{M}$ into a single-objective MDP $\mathcal{M}'$ with the scalarised reward function $lu\dot{R}$. Of course, all optimal policies of the single-objective MDP $\mathcal{M}'$ are precisely the $lu$-optimal policies of $\mathcal{M}$ (for the technical proof of this see Section 2.2 of the main paper).

(iii) These two facts together tell us: given a utility function $f$ that can be decomposed into a strictly monotonically increasing function $smi$, and a linear function $lu$, then $f(x) = smi(lu(x))$ will have deterministic and stationary $f$-optimal policies (which will be exactly the deterministic and stationary $lu$-optimal policies).

Next, we prove that the following two functions are quasi-representative: a function decomposable in an affine function and a strictly monotonically increasing function, and another decomposable in a linear function and a strictly monotonically increasing function.

(iv) Next, Lemma 8 proves that any affine utility function $af$ is quasi-representative of another linear utility function $lu$. More precisely, in every MOMDP, all $af$-optimal policies are also $lu$-optimal policies and vice-versa for the linear utility function $lu$ defined as $lu(x) = af(x) - af(0)$. This is because any affine function $af(x)$ can be decomposed as $af(x) = A(x) + b$, with $A(x)$ being a linear function and $b = af(0)$, a constant.

(v) Now, consider a utility function $f(x) = smi(af(x))$ that can be decomposed as a product of a strictly monotonically increasing utility $smi$ function and an affine utility function $af$. Consider also the utility function $f'(x) = smi(af(x) - af(0))$. This second function $f'(x)$ is composed by a strictly monotonically increasing function and a linear function, so as previously proven in $(iii)$, there are deterministic and stationary $f'$-optimal policies.

(vi) Then, recall that by $(iv)$, utility functions $af(x)$ and $af(x) - af(0)$ are quasi-representative (i.e., they share the same optimal policies). Thus, it is clear that $smi(af(x))$ and $smi(af(x) - af(0))$ are also quasi-representative, because strictly monotonically increasing functions preserve the ordering by definition.

Finally, since $f'(x) = smi(af(x) - af(0))$ has deterministic and stationary $f'$-optimal policies, and $f(x) = smi(af(x))$ and $f'(x)$ are quasi-representative, we conclude that there are also deterministic and stationary $f$-optimal policies. $\qquad\square$

