# OpenReview forum: "An Analytical Study of Utility Functions in Multi-Objective Reinforcement Learning"
_NeurIPS.cc/2024/Conference — NeurIPS 2024 poster_

### Official Review · Reviewer_59cy · 2024-06-17

**Soundness:** 4
**Presentation:** 3
**Contribution:** 3
**Rating:** 7
**Confidence:** 4

**Summary:**

The paper considers the problem of multi-objective RL and performs a rigorous theoretical analysis on the space of optimal policies as a function of the utility functions and preferences. Specifically, in teh case of utility optimal policies, it considers two types of optimality - at the state level and at all states level. The key contribution in this sub-task of utility optimality lies in demonstrating that the more common utility function conditions such as monotonicity, differentiability and strictly montonically increasing and continuously differentiable are all insufficient to guarantee optimality. Simple counterexamples are provided for each of these cases to prove their insuffiency.

Given these observations, a signficiant contribution of the paper is in the identification of the conditions under which utility optimal policies exist for all states. Specifically, the paper identifies that a decomposable utility function of the form h(g(x)) where g is an affine function (linear with a constant) is one such condition under which the u-optimal policy exists. The second key contribution of the paper lies in teh fact that it considers user preferences, and identifies quasi-representative preference relations allows us to identify u-optimality.

**Strengths:**

+ The paper is written well. It idenfies the problems clearly, motivates them well and provides simple counterexamples.

+ The problem addressed in this paper is an important one, that of deeply understanding multiobjective RL. It is probably the most natural setting inside RL and not enough attention is being given to this challenging task. So from that perspective, this is a very interesting read. I thank the authors for carefully constructing the problem and clearly explaning the challenges and then present their observations.

+ The two observations of decomposable utility (with affine functions) and the quasi-representative preferences are quite important.

**Weaknesses:**

- While the paper is written well and the problems are motivated well, I would have liked to see a specific discussion on the types of settings/problems where such situations are plausible/common. For example, where would one observe this decomposable affine functions or quasi-representative preferences in real-world? It would be nice to see a section with some real examples to make this paper's analyses clearer.

- While I understand that the having a decomposable utility (with affine functions) can result in u-optimality, what is unclear is that are they sufficent or complete conditions?

- Same issue with the quasi-representative preferences. While they themselves can be complete and transitive, is the condition of quasi-representativeness sufficient for optimality? If so, can you expand how?

**Questions:**

Please see the weakness part. I specifically would like to understannd the sufficiency and completeness of these conditions and if possible a discussion of the situations where these are both practical or common.

**Limitations:**

There are no significant potential societal negative impacts.

---

> ### Author Rebuttal · Authors · 2024-08-02
>
> We thank the reviewer for their comments, suggestions, and feedback. We appreciate that the reviewer agrees with us in the need for addressing in more depth the theoretical aspects of multi-objective reinforcement learning.
>
> We proceed to answer the questions:
>
> **Question 1:
>
> Regarding the types of problems that we address and their link to real-world problems. We cannot enter into much detail here, but linear utility functions (a particular kind of affine functions) represent a large majority of utility functions in real-world applications (due to their simplicity and well-known properties). To give one recent example,  Rodriguez-Soto et al. in "Multi-objective reinforcement learning for designing ethical multi-agent environments" (2023) showed an example gathering scenario in which agents needed to face lexicographically two objectives (one individual related to how much the agent can gather for itself, and another one ethical, related to how much it helps the community), and provided a linear utility function that was able to reach lexicographic solutions (i.e., agents learned to prioritise the ethical objective over the individual one).
>
> We will also mention in our camera-ready version some of the practical examples of utility functions in the most recent MORL surveys, such as Hayes et al. "A practical guide to multi‑objective reinforcement learning and planning" (2022). These examples tackle problems route planning, water management, and wind farm control.
>
> Interestingly, another large body of work in MORL has tackled the problem of assuming an unknown utility function and trying to compute a solution set general enough to at least include a solution to this unknown function. This may lead to unexpected problems, as we have tried to show in this paper, in which we put the focus back into the utility functions themselves.
>
> Regarding preferences for which a quasi-representative utility function exists. As mentioned in the paper, the first two conditions of our Theorem 3 (Completeness and transitivity) are very common conditions for preference relations in game theory, so we expect it to cover the majority of most used preference relations. The third condition of Theorem 3 (that a maximal element exists per state) is virtually assumed by the whole MORL community since the goal is to always maximise the utility function (implicitly assuming that there exists a maximal element).
>
> In fact, it is difficult for us to even think of an example preference relation for which a quasi-representative utility function would exist without satisfying the conditions of Theorem 3. And recall from our definition that if the utility function is not quasi-representative of the preference relation, then maximising the utility would lead to not maximising the preference relation.
>
>
> **Question 2:
>
> We apologise if Theorem 2 was not clear enough. The conditions of Theorem 2 are only sufficient conditions, but not necessary. As a quick example, notice that for any single-state MOMDP, any <u,s>-optimal policy of this single state is also by definition a <u>-optimal policy.  Theorem 1 proves that we only need continuity to reach a <u,s>-optimal state (much more relaxed condition than being affine).
> In future work we expect to further study the necessary conditions for a <u>-optimal policy to exist in general MOMDPs.
>
>
> **Question 3:
>
> We hope that we are understanding the reviewer properly on this question, if not, we will correct our answer in the reviewer-author discussion.
>
> Again, we apologise if Theorem 3 was not clear enough. Theorem 3 only provides the sufficient conditions that a preference relation needs to meet to guarantee that an associated quasi-representative utility function exists.
>
> However, the reviewer is right on their intuition: it is sufficient for a utility function to be quasi-representative of a preference relation in a given MOMDP to guarantee that a <u,s>-optimal policy exists for every state of the MOMDP. We did not have space in the final paper to include this result.
>
> It is easy to prove: Theorem 3 requires the preference relation to have at least one maximal element (policy) per state. Then, if the utility function is quasi-representative of this specific preference relation, we require it to be the maximum element (policy) per state. In other words, we require it to have a <u,s>-optimal policy per state.
>
> Again, the necessary conditions for both issues here are future work that we are eager to tackle.

---

> > ### Comment · Reviewer_59cy · 2024-08-09
> > **Thanks for the response**
> >
> > I thank the authors for taking the time to carefully read the review and write the response.
> >
> > My questions on the two theorems were about sufficency vs necessary conditions. The response has made this quite clear to me. I hope that the authors will make the theorems clearer in the next iteration of the paper.
> >
> > Finally, there were some more multiobjective learning papers in mid 2000s (they were called as multicriteria RL back then). I suggest that the authors cite those papers in their paper as well for the sake of completion.

---

### Official Review · Reviewer_WWNq · 2024-06-18

**Soundness:** 3
**Presentation:** 3
**Contribution:** 3
**Rating:** 6
**Confidence:** 4

**Summary:**

This paper studies the expressiveness of utility functions in multi-objective reinforcement learning (MORL). In MORL, the reward function of the MDP is a vector of multiple (possibly) conflicting reward functions, and the goal of an agent is to maximize a given utility function (function mapping the vector value function to a scalar) that expresses the preferences of a user. In particular, the paper studies two problems within MORL: (i) which utility functions are guaranteed to have an associated optimal policy? (ii) whch user preferences can be expressed via some utility function? The paper shows that, for (i), a continuous utility function is enough for optimality in a given state, and utility functions decomposable as a combination of an affine and a strictly monotonically increasing function is enough for optimality in all states. Regarding (ii), the authors show that for quasi-representative preference relations that satisfy a few conditions (e.g., completeness, transitivity), a quasi-representative utility function always exists that expresses the given preference.

**Strengths:**

- The problem studied in this paper is very relevant to the MORL community.
- The paper is written clearly and the results are easy to follow. The authors present examples that make it easier to understand the introduced theorems.
- The distinguishment between “preferences” and “utility functions” is, to the best of my knowledge, a novel perspective that can help the study of MORL under non-linear preferences.

**Weaknesses:**

- The main weakness of the paper is that it does not do a good job in discussing previous theoretical works in MORL and comparing/discussing its findings with the related literature (see below).
- A few introduced results are not entirely novel but phrased differently than in previous works. For instance, that a stationary deterministic optimal policy may not exist for non-linear utility functions is a known result (Example 3).
- Although the paper showed sufficient conditions for preferences being possible to express in terms of utility functions, it does not discuss how the results could be used to create or solve such utility functions in practice.

**Questions:**

In general, the paper introduces a few interesting and potentially useful theoretical results for MORL. However, the paper lacks a more in-depth discussion of how these results can be applied to design novel algorithms. This, combined with the fact that the ideas are not discussed in light of many previous results, makes the paper not yet ready for publication.
Below, I have a few questions and constructive feedback to the authors:

1) My main concern is that the paper does not have a related work section and does not discuss many relevant related works. For instance:

[1] showed how to construct stochastic policies that are optimal w.r.t. the initial state distribution.

[2] introduced different solution sets that extend the definition of the Pareto front and can express more utility functions.

[3] and [4] showed that many multi-objective utility functions can not be expressed via Markov rewards.

[5] discusses the expressivity of many RL formalisms to represent different preferences, including MORL.

2) Regarding the examples, e.g., Example 3, it would be much clearer and easy to read if the authors provided an image of the MDP as graph.

3) Theorem 1 seems to be a more formal definition of the result in Vamplew et al. 2009 [1].

4) Proof of Theorem 2 looks incomplete. Lemmas 7 and 8 consider single-objective MDPs, not MOMDPs. It is not clear how they can be combined to prove Theorem 2.

5) Theorem 2 shows one class of non-linear utility functions that have a deterministic stationary optimal policy. However, is it possible that a more general class exists? I would suggest adding this discussion.

Minor:
- Use large brackets, e..g., in Equation 1, the bracket is smaller than the summation symbol.
- In definition 3, the transition function is $\mathcal{T}$ instead of $T$. Please be consistent.

[1] Vamplew, P., Dazeley, R., Barker, E., and Kelarev, A. (2009). Constructing stochastic mixture policies for episodic multiobjective reinforcement learning tasks. In AI 2009: Advances in Artificial Intelligence.

[2] Röpke, W., Hayes, C. F., Mannion, P., Howley, E., Nowé, A., and Roijers, D. M. (2023). Dis-
tributional multi-objective decision making. In Elkind, E., editor, Proceedings of the Thirty-Second International Joint Conference on Artificial Intelligence, IJCAI-23.

[3] Skalse, J. and Abate, A. (2023). On the limitations of Markovian rewards to express multi-objective, risk-sensitive, and modal tasks. In Evans, R. J. and Shpitser, I., editors, Proceedings of the Thirty-Ninth Conference on Uncertainty in Artificial Intelligence.

[4] Miura, S. (2022). On the expressivity of multidimensional markov reward. In Proceedings of the
Conference on Reinforcement Learning and Decision Making.

[5] Subramani, R., Williams, M., Heitmann, M., Holm, H., Griffin, C., & Skalse, J. (2023). On The Expressivity of Objective-Specification Formalisms in Reinforcement Learning. ICLR 2024.

**Limitations:**

The results have some limitations that should be addressed (see Questions above). The paper has no potential negative societal impact.

---

> ### Author Rebuttal · Authors · 2024-08-02
>
> We thank the reviewer for their insights and suggestions.
>
> **Weakness 2:
>
> Yes, Example 3 illustrates a well-known fact in the MORL literature. In case of acceptance, we will state it. However, we wanted to include it before showing Example 4, which builds on Example 3 by providing an example of both strictly monotonically increasing and continuously differentiable function for which no u-optimal policy exists, a completely novel result.
>
> **Weakness 3:
>
> We accept it, but we believe the topic of classifying the conditions for the existence of solutions, due to its novelty in the MORL community, deserved the entire paper (in fact, we had to struggle to fit in all the main results). Solving MORL problems is out of the scope of this paper.
>
> **Question 1:
>
> While the recommended papers tackle relevant MORL problems, papers [1, 2, 4, 5] aim to solve a different problem than ours. Paper [3] is the most similar to ours, but they make no distinction between <u,s>-optimal and <u>-optimal policies. This is a major difference that we will discuss in the following paragraphs.
>
> Nevertheless, we do agree with the reviewer that our paper would benefit from an expanded related work section, which is currently compressed in Section 2 due to space limitations. We will include comments explaining the similarities and differences to these 5 papers upon acceptance.
>
> Let us clarify how these papers relate to ours:
>
> - Papers [1] and [2] tackle a different problem: they focus on creating novel solution concepts in MORL and methods to solve them. Instead, we focus on characterising for which families of utility functions this solution exists. In [1], they present a method for computing the Pareto front (PF) from a given state. They implicitly assume that the PF will include the <u,s>-optimal policy, but our Example 5 proves that this is not always the case.
>
> - Paper [4] tackles a complementary, though also different, problem to ours. They define preferences as sets of “acceptable policies' ' and aim to find for which environments they can set the constraints and reward functions of a constrained MDP (CMDP) for which the acceptable policies are optimal. CMDPs and MOMDPs are similar but separate research areas, being the major difference that a MOMDP does not have constraints, but a scalarization function.
>
> - Paper [3], which we were not aware of, slightly overlaps with ours. Their theoretical results complement ours by stating that, for every “objective” (preorder between policies), a <u>-optimal policy exists if and only if this objective can be represented with a linear utility function. This aligns with our results with Theorem 2. However, they do not establish whether there may be more families of utility functions for which a u-optimal policy exists, as we do with Theorem 2. Moreover, our “preference” definition allows for ordering policies in each state of the environment, providing more granularity than their “objective” definition. This difference is also significant, because it allows us to identify issues in the solution concepts of MORL as we have tried to illustrate with our examples. As a side note, we see their Corollary 2 reaches the same conclusion as us in Example 8. We will rewrite Example 8 to recognise their work.
>
> - Finally, Paper [5] follows on [3], but tackles a different problem than us. They compare the expressivity of the MORL framework with other frameworks. They aim to know which objectives can be represented on each framework. But here we tackle a different problem: we focus strictly on MOMDPs, and aim to clarify for which utility functions it makes sense to compute a u-optimal policy. Moreover, like [3], their “objective” definition does not allow them to order policies differently per state, unlike our “preference” definition.
>
> **Question 2:
>
> Due to page limitations we were not able to include figures. In case of acceptance, we will try to fit as many supporting figures to the examples as we can.
>
> **Question 3:
>
> No, Theorem 1 is not a more formal definition of the result in [1]. These are unrelated results. While [1] provides a method for computing <u,s>-optimal policies for monotonically increasing utility functions, we prove that a <u,s>-optimal policy always exists for any continuous utility function.
>
> Vamplew et al. proved that a Pareto front can be constructed from the deterministic policies of a convex coverage set (CCS). In our terms, they state that the <u,s>-optimal policy (if it exists) of any monotonically increasing utility function can be computed by first computing a CCS for state s.
>
> Meanwhile, our Theorem 1 states that for any continuous utility function, there exists a <u,s>-optimal policy. Our proof relies on the fact that all possible value vectors (at a given state) are contained in the CCS. Notice that our Example 5 is a counter-example to the methodology of [1]. Following their methodology we would not have found any <u,s>-optimal policy for the utility function of Example 5, because no <u,s>-optimal policy exists for it. This example remarks the importance of formalising <u,s>-optimal policies.
>
> **Question 4:
>
> Sorry for the typo. Lemma 8 should read “For every finite multi-objective MDP, […]”. We hope that now this is clearer.
>
> **Question 5:
>
> Indeed, this is a very interesting question! We hope to study this problem in future work, because we can think of example utility functions that do not satisfy Theorem 2 conditions while also having a u-optimal policy in some MOMDPs. The theoretical implications of Example 4 make it difficult finding more families of utility functions for which a u-optimal policy always exists. We know that if another family of optimally solvable utility functions exists, they need to satisfy the Bellman optimality equation. Finding this alternative family of utility functions would be an impactful contribution to the MORL community.
>
> **Minor questions:
>
> In case of acceptance we will make sure of correcting all typos.

---

> > ### Comment · Reviewer_WWNq · 2024-08-09
> > **Response**
> >
> > I thank the authors for their careful response to my questions and concerns. I am slightly increasing my score due to the authors' response. Below, I have a few more comments on the points raised by the authors:
> >
> > **Weakness 3: I agree that algorithms for solving MORL problems are out of the scope of the paper. However, since the end goal of the field is the design of MORL algorithms, I strongly suggest adding a short discussion to the paper on how these theoretical ideas might be useful. E.g., "Theorem X implies that the community need to design novel algorithms with properties Y and Z.".
> >
> > **Question 1: Thank you for providing this comparison. It is critical that these discussions are included in the final version of the paper, since they explain why previous related works are not sufficient.
> >
> > **Question 4: I believe this is not the only issue in this proof. Please provide more detailed proof of Theorem 2, which currently is "Direct consequence of combining Lemma 7 and Lemma 8". It is very unclear how this is the case.

---

> ### Author Response · Authors · 2024-08-09
>
> We agree with reviewer's point regarding the importance of designing MORL algorithms in the field. In case of acceptance or in any future iteration of the paper, we will definitely add both a discussion section of how our theorems impact algorithm design, and a related work section, following the reviewer's suggestion.
>
>
> We will now provide an-indepth proof of Theorem 2. In case of acceptance we will make public this proof together with all appendix material.
>
> (i) - First, as Lemma 7 states, applying a strictly monotonically increasing utility function to a single-objective MDP does not modify its set of deterministic and stationary optimal policies.
>
> (ii) - Second, every linear utility function "lu" can transform a MOMDP M into a single-objective MDP M' with the scalarised reward function "lu \dot \vec{R}". Of course, all optimal policies of the single-objective MDP M' are precisely the lu-optimal policies of M (for the technical proof of this please see Section 2.2 of our paper).
>
> (iii) - These two facts together (i)+(ii) tell us: given a utility function "f" that can be decomposed into a strictly monotonically increasing function "smi", and a linear function "lu", then "f(x) = smi(lu(x))" will have deterministic and stationary f-optimal policies (which will be the deterministic and stationary lu-optimal policies).
>
> (iv) - Next, Lemma 8 proves that any affine utility function "af" is quasi-representative of another linear utility function "lu".  More precisely, in every MOMDP,  all af-optimal policies are also lu-optimal policies and vice-versa for the linear utility function "lu" defined as lu(x) = af(x) - af(0). This is because any affine function af(x) can be decomposed as  "af(x) = A(x) + b", with A(x) being a linear function and b = af(0), a constant.
>
> Now, consider an utility function "f(x) = smi(af(x))" that can be decomposed as a product of a strictly monotonically increasing utility "smi" function and an affine utility function "af".
>
> (v) - Consider also the utility function "f'(x) = smi(af(x)-af(0))". This second function f'(x) is composed by a strictly monotonically increasing function and a linear function, so by (iii), there are deterministic and stationary f'-optimal policies.
>
> (vi) -  Then, recall that,  by (iv), utility functions "af(x)" and "af(x)-af(0)" are quasi-representative (i.e., they share the same optimal policies). Thus, it is clear that "smi(af(x))" and "smi(af(x)-af(0))" are also quasi-representative, because strictly monotonically increasing functions preserve the ordering by definition. This is a point that should have been much clearer in the paper, so we apologise for that.
>
> Finally, since:  f'(x) = smi(af(x)-af(0)) has deterministic and stationary f'-optimal policies, and f(x)= smi(af(x)) and f'(x) are quasi-representative, we conclude that there are also deterministic and stationary f-optimal policies. This proves Theorem 2.
>
> If there is any further doubt or unclear proof we will be happy to clarify it.

---

> > ### Comment · Reviewer_WWNq · 2024-08-12
> >
> > Thank you for clarifying the proof.
> >
> > I have one last question regarding Theorem 2: You showed a class of utility functions with deterministic and stationary optimal policies. However, because of the inner affine function, it seems that this utility function can represent the same set of optimal policies as linear utility functions. In other words, given a utility function "u" decomposable as a strictly monotonically increasing function and an affine function, its optimal policy will be on the Convex Hull (Eq. 5). Is that correct? If so, why would this class of utility function be useful if it can represent the same policies as linear utility functions? If this is not correct, can you provide an example where this utility function has an optimal policy whose value is a concave region of the Pareto front/undominated set?

---

> ### Author Response · Authors · 2024-08-12
>
> This is correct: Theorem 2 proves that any function of such class is quasi-representative to another linear utility function, and thus its optimal policies belong to the convex hull.
>
> The usefulness of this class of utility functions is threefold, in our opinion:
>
> 1) it proves that not only linear utility functions have solution policies inside the convex hull.
>
> 2) similarly, it provides more structure to the convex hull, allowing us to identify a particular family of utility functions that belong to it.
>
> 3) In practice, Theorem 2 provides a general methodology for obtaining solution policies: if a utility function can be proved to be quasi-representative to a linear utility function, then an u-optimal policy is guaranteed to exist. Take for instance the utility function of Example 6. A priori, one would not know how to compute its u-optimal policy (or if it even exists). Thanks to Theorem 2 we know that it exists, and that it belongs to the convex hull.
>
> To understand the significance of these results, we would also like to remark that:  it is still a complex problem to find utility functions for which an u-optimal policy always exists for any arbitrary MOMDP, and that furthermore this u-optimal policy does not belong to the convex hull. We are not aware of a single utility function for which this is true.
>
> Moreover, the theoretical implications of Example 4 greatly limit the   candidate families of utility functions for which a deterministic and stationary u-optimal policy is guaranteed to exist.
>
> In any case, we already started working on searching for utility functions with concave deterministic and stationary u-optimal policies, and hope to find results in future work.

---

> > ### Comment · Reviewer_WWNq · 2024-08-12
> >
> > Again, I thank the authors for their insightful response. I am increasing my score since the theoretical results in the paper can be potentially useful to other researchers in the field.

---

### Official Review · Reviewer_nkkJ · 2024-07-12

**Soundness:** 2
**Presentation:** 3
**Contribution:** 2
**Rating:** 4
**Confidence:** 3

**Summary:**

The authors studied preference relations and utility functions, which are the main components of utility-based MORL. Many prior works assumed two things: 1) for a given preference, there exists a utility that captures the preference, and 2) for a given utility, there exists an optimal policy. The authors provide several counterexamples for these assumptions and suggest sufficient conditions for each assumption to be true.

**Strengths:**

-	The motivation of the study is important in utility-based MORL.
-	The authors provided several examples to show that a representative utility and an optimal policy may not always exist. These examples help to understand the motivation of this work.

**Weaknesses:**

- The suggested sufficient conditions do not seem that surprising and are quite straightforward. In particular, Theorem 2 appears to depend directly on prior results for affine utility and strict monotonicity.
- To use Theorem 3, the relation must be a total order. In this setting, the Pareto dominance relation (x⪰Py iff xi≥yi∀i and  xi>yi∃i), which is widely used in MORL, is not applicable since it is a partial order.
- Section 3 and 4 seem misaligned. (please see question 3)

**Questions:**

- Is there any interpretation for how extensively the suggested sufficient conditions cover the set of desired preferences or utilities? For example, in Theorem 1, I understand that continuous utility has a <u,s>-optimal policy, but isn't this class too small to have a <u,s>-optimal policy?
- Theorem 2 gives a sufficient condition for a deterministic stationary <u>-optimal policy. As I understand it, this condition guarantees the same property as in single-objective MDPs: the existence of a stationary deterministic optimal policy. However, in MORL, it is common for a stationary deterministic optimal policy not to exist. Is there any sufficient condition for a stationary stochastic <u>-optimal policy (even when there is no stationary deterministic optimal policy)? I guess that this result would be more helpful for MORL field.
- In my understanding, the constructed u in the proof of Theorem 3 is discontinuous. If this is true, then even a <u,s>-optimal policy is not guaranteed to exist according to the Theorem 1.
- (minor) In section 4, ⪰ is used for both vector preference (V(s)) and function preference (V). I guess that these need to be distinguished in notation.

**Limitations:**

The motivation and importance of this work are appealing. However, the sufficient conditions seem to result in a too small subclass of interest.

---

> ### Author Rebuttal · Authors · 2024-08-01
>
> We thank the reviewer for the provided suggestions and comments.
>
> We hope that our answers will clarify the soundness of our paper. Reviewer's rating of 3 indicates “technical flaws, weak evaluation, or inadequate reproducibility”. We consider that none of them are the case in our paper. We will rewrite the unclear parts of the paper in case of acceptance to further clarify all technical doubts.
>
> ** Weakness 1:
>
> Yes, Theorem 2 directly depends on previously well-known results in the MORL literature, though no one has connected them yet. Notice however that while Theorem 2 provides sufficient conditions, we also provide non-trivial proofs of insufficient conditions that complement Theorem 2 results, and other two theorems in the paper that tackle completely new topics in MORL:
>
> - Theorem 1 is the first result in MORL that characterizes when a solution policy exists in MORL for a single state (despite the concept has been used experimentally since the creation of MORL).
> - Theorem 3 is the first result in MORL that characterizes when a preference relation can be represented as a utility function (despite preference relations being used since the creation of MORL).
> - Example 4 proves that strictly monotonically increasing utility functions (despite their appeal in the MORL community) do not necessarily have a solution policy for any state (<u,s>-optimal policy in our terms). We are certain that this will be a shocking result in the community that will need to think again on their focus on monotonically increasing utility functions.
> - Example 5 proves that both strictly monotonically increasing and continuously differentiable (again, some of the most popular families of utility functions in the MORL community) do not necessarily have a global solution policy (u-optimal policy in our terms).
>
> In summary, all these results lead us to conclude that u-optimal policies are very difficult to guarantee in general, a problem that the MORL community was still not aware of.
>
> **Weakness 2:
>
> We argue that such weakness does not hold for the following reasons:
>
> - It is right that Theorem 3 requires a total order among policies for every state s. The reviewer is also right that Pareto-dominance only imposes a partial order among policies.
>
> - But Pareto-dominance is a “solution concept” in MORL, not a utiltiy function, nor a family of functions. Roijers and Whiteson in their textbook “Multi Objective Decision Making” [16] provide a very good explanation of the difference.
>
> - This distinction is important here, because every utility function imposes a total order among policies for every state. Proof: for every state s, the scalarized value of every policy  V^{\pi}(s) will be a scalar number. Thus, we can totally order all policies at state s.
>
> Hence, Theorem 3 applies to every possible utility function (even monotonically increasing ones, whose solution lies on the Pareto front), contrarily to what was said in weakness 2.
>
> **Question 1:
>
> Regarding the extensiveness of the sufficient conditions of our Theorems. Recall that all theorems refer to finite MOMDPs, which are the most widely used kind of MOMDPs.
>
> - Then, Theorem 1 only demands the utility function to be continuous to guarantee the existence of <u,s>-optimal policies. Continuous utility functions are the most extensively known and used family of functions, including in MORL. Notice that in particular linear utility functions, widely used in MORL, are continuous.
>
> - Theorem 2, despite its apparent simplicity, already covers one of the most “desired” utility functions: linear utility functions. Again, as previously mentioned, Theorem 2 should be observed together with Example 5, which also covers families of desired utility functions (monotonic ones) but providing negative results for them.
>
> - Finally, as previously explained, Theorem 3 applies to every utility function. It does not apply to every possible preference relation, but our conditions copy standard conditions in game theory, which we consider to be representative enough.
>
> ** Question 2:
>
> Finding deterministic and stationary solution policies is very relevant and a major research topic for  MORL research. Following Roijers and Whiteson’s classification of MORL solution concepts in [16], half of them involve the computation of deterministic and stationary policies, which our Theorem 2 directly addresses.
>
> Regarding the reviewer’s concern about expanding our result to stochastic policies: This is future work for us. We have tried to find such kind of families, but so far, we have only found negative results (e.g., Example 5) that greatly reduce the space of viable policies for which a stochastic u-optimal policy would always exist. However, we consider that these negative results will also be helpful for the MORL community.
>
>
> ** Question 3:
>
> While Theorem 1 and 3 are related results, they do not directly affect each other. Probably this was not clear enough on our part.
>
> - The constructive proof of Theorem 3 provides a discontinuous utility function. Theorem 1 guarantees that any continuous utility function will have a <u,s>-optimal policy, but does not deny that some discontinuous optimal policies may have a <u,s>-optimal policy.
>
> - Notice that having a <u,s>-optimal policy for some state s means that there is a maximum element among scalarized values of policies at state s. The utility function of Theorem 3 is specifically constructed to always have a maximum element for each state, thus guaranteeing the existence of <u,s>-optimal policies by definition.
>
> - In summary, the reviewer's inference does not hold: even though the utility function of the proof of Theorem 3 discontinuous, it is guaranteed to have a <u,s>-optimal policy.
>
>
> **Question 4:
>
> We agree with the reviewer that this was an abuse of notation from our part. We will correct it in case of acceptance.

---

> > ### Comment · Reviewer_nkkJ · 2024-08-13
> >
> > Thanks for the reply. I have raised my rating.

---

### Decision · Program_Chairs · 2024-09-25

**Decision:**

Accept (poster)

**Comment:**

This paper investigates preference relations and utility functions—critical components of utility-based multiobjective RL (MORL) methods. In particular, the authors presented several counterexamples for key assumptions made in previous works and discussed sufficient conditions for each assumption to hold. They also provided several examples to show that a representative utility function and an optimal policy may not always exist.

Reviewer nkkJ raised a few concerns, including, e.g., that Thm. 2 appears to directly depend on/extend well-known results in the MORL literature. The authors responded by pointing out that while this is true, up to this point, no one had made the connections presented in this paper. They also argued that while Thm. 2, on its own, does provide sufficient conditions, the current paper *additionally* provides non-trivial proofs of *insufficient* conditions that complement this theorem. They also argued that Thm. 1 is the first result of its type in the MORL literature. Post-rebuttal, nkkJ increased their recommendation. Overall, I agree with the authors' point of view in this case: a borderline reject vote (nkkJ's recommendation) often suggests that the paper may have technical flaws, weak evaluation, or inadequate reproducibility. I do not believe this is the case. I believe (and other reviewers share this opinion) that the examples provided in this work may give valuable insights to the MORL community.

Reviewer WWNq brought up a few other concerns, some similar to those nkkJ mentioned; e.g., that some results presented in this paper are only partially novel. As an example, it is well-known that a stationary deterministic optimal policy may not exist under non-linear utility functions. This reviewer's main concern is that the paper did not have a related work section and did not adequately discuss many relevant related works. Regarding this point, WWNq listed five papers that present theoretical findings similar to those discussed in this work. After a fruitful discussion with the authors, the reviewer chose to increase their score since "(...) the theoretical results in the paper can be potentially useful to other researchers in the field". Finally, reviewer 59cy mentioned a few potential limitations of this work but was overall satisfied with the paper's contributions and the authors' responses.

Although the consensus among the reviewers is that the ideas presented in this work are not entirely novel, it is clear that the authors successfully made important connections that had yet to be clearly discussed in the literature. Papers of this type are a valuable resource to the scientific community since they help consolidate in a centralized way a series of key insights that are meaningful and helpful to other researchers in the field. For this reason, I believe that the NeurIPS community may benefit from the discussion and insights presented in this work. The reviewers strongly encouraged the authors to address the concerns and suggestions that were brought up during the discussion phase.